# *Apophysomyces jiangsuensis* sp. nov., a Salt Tolerant and Phosphate-Solubilizing Fungus from the Tidelands of Jiangsu Province of China

**DOI:** 10.3390/microorganisms8121868

**Published:** 2020-11-26

**Authors:** Siyu Li, Ruiming Han, Huanshi Zhang, Yongchun Song, Fugeng Zhao, Pei Qin

**Affiliations:** 1Halophyte Research Lab, Nanjing University, Nanjing 210008, China; 32029@qzc.edu.cn (S.L.); ruiming.han@njnu.edu.cn (R.H.); fgzhao@nju.edu.cn (F.Z.); 2Qu Jiang Experimental Middle School, Quzhou 324022, China; 3Jiangsu Center for Collaborative Innovation in Geographical Information Resource Development and Application, Jiangsu Key Laboratory of Environmental Change and Ecological Construction, School of Environment, Nanjing Normal University, Nanjing 210023, China; 4Nanjing Institute for Comprehensive Utilization of Wild Plants, Nanjing 210042, China; zhanghuanshi@126.com; 5Institute of Functional Biomolecules, Nanjing University, Nanjing 210093, China; songych@nju.edu.cn

**Keywords:** *Apophysomyces*, taxonomy, phosphorus-dissolving, soil fungi

## Abstract

A newly isolated phosphate-solubilizing fungus from the topsoil of *Spartina alterniflora* habitats in Yancheng coastal salt marsh was cultivated. Scanning electron microscopy observation revealed that the sporangia are nearly spherical, peach-shaped, and the spores formed on the top of sporangia. The spores are ellipsoidal with raised white nubbins on the surface. Based on a polyphasic study and the genetic distance analysis referring to the sequence analysis of ITS (ITS1 + 5.8S + ITS2) and 28S rDNA (D1/D2 domains) genes, the novel species belongs to the genus *Apophysomyces* and is named as *A. jiangsuensis*. The optimum growth temperature and salinity of the new species were 28 °C and 1.15% NaCl, respectively. A study of its phosphate-solubilizing ability revealed that the fungus had an obvious decomposition effect on lecithin, Ca_3_(PO_4_)_2_, and AlPO_3_, respectively. The pH of the fermented liquid progressively decreased from 6.85 to 2.27 after 7 days of incubation, indicating that the low molecular weight organic acids excreted into the culture liquor were oxalic, succinic, and malic acids and a trace amount of citric acid. Among these, oxalic acid was the major organic acid, and its amount reached 652.5 mg/L. These results indicated that the main mechanism underlying the dissolved phosphorus was related to the secretion of large amounts of organic acids.

## 1. Introduction

The genus *Apophysomyces* was first proposed by Misra et al. (1979) and *A. elegans* was the type specimen that was isolated and identified in the soils in Northern India [1]. *Apophysomyces* belongs to the order Mucorales and Benny et al. (2001) identified it as a genus in the family Mucoraceae based predominantly on morphology [2], while Hoffmann et al. (2013) put it into Saksenaeaceae based on a phylogenetic analysis of four molecular markers [3]. Meanwhile, Wijayawardene et al. (2020) confirmed that *Apophysomyces* was contained in Saksenaeaceae on the basis of phylogeny of the kingdom Fungi [4].

The characteristics of the type species *A. elegans* include pyriform sporangia, conspicuous funnel- and/or bell-shaped apophyses, and subhyaline, thin-, smooth-walled, and oblong sporangiospores that terminate in a columellate, multispored sporangium [1]. Additionally, it is pretty thermotolerant that can grow rapidly between 26 and 42 °C [5]. *Apophysomyces* is usually found in soil, decaying vegetation, and detritus, which has been reported to cause severe human infections that is more common in tropical and subtropical regions [1,6]. The type species *A. elegans* has been reported as an agent of zygomycosis in immunocompromised patients [7,8]. The method of combining molecular biology and morphological characteristics was used by Alvarez et al. (2010), and three species were identified, namely, *A. ossiformis*, *A. trapeziformis*, and *A. variabilis* [9]. *A. mexicanus* was isolated from an infected car accident patient and shown to have caused the disease, according to Bonifaz et al. [10]. *A. thailandensis*, which can solubilize metal minerals, was isolated from soil in Chiang Mai Province, Thailand [11]. Six species of the *Apophysomyces* have been discovered thus far, all of which failed to sporulate on routine mycological media [9,10,11].

Soil phosphorus is among the crucial limiting factors in agricultural practices. About 43% of the world’s 1.319 billion ha of arable land is phosphorus-deficient [12]. Phosphate-solubilizing microorganisms (PSMs) refer to the functional group of microorganisms that can transform insoluble soil phosphate into soluble phosphorus in the course of their living activities [13,14]. PSMs dissolve poorly soluble mineral phosphates in soil through various processes, such as acidification, proton exchange, chelation, and enzymatic hydrolysis [15]. The efficiency to solubilize rock phosphate (RP) by *Trichoderma* and *Aspergillus* strains depends on the release of organic acids with low acidity constants and it is irrelevant to the concentration when they are released. It can be raised that the production of organic acids is the main mechanism of RP dissolution [16]. In our previous work, a formula derived from the mixture of arbuscular mycorrhizal fungi (*Glomus mosseae*) and mucoroid fungus (*A. jiangsuensis*) proved that a valid P-solubilizing agent could increase the content of available phosphorus in saline lands. Results showed that certain fungal strains could be promising tools to improve the utilization of saline soil phosphorus resources [17]. Further investigation revealed that although AM fungal hyphae were hampered in terms of colonization capacity, spore germination, and growth by salinity, NaCl promoted certain PSM (*A. jiangsuensis*) populations in the rhizosphere of saline crops, and PSM abundance was correlated positively with plant growth and shoot phosphorus concentration [18]. Considering the extreme importance of P nutrition in saline agriculture, it is important to isolate and identify cryptic salt-tolerant P-solubilizing fungi species.

The objectives of present work were as follows: (i) to describe salt tolerant P-solubilizing fungus *Apophysomyces jiangsuensis* as a new species, isolated from the topsoil of *S. alterniflora* habitats in Jiangsu tideland, China, based on the results of a phylogenetic study and the genetic distance analysis referring to the sequence analysis of ITS and 28S rDNA genes, and to evaluate its morphological characteristics; (ii) to supply a method to stimulate the production of spores; and (iii) to study the phosphate-solubilizing ability and preliminarily explore the P-solubilizing mechanisms of the new species.

## 2. Materials and Methods

### 2.1. Fungal Strain

The fungal strain studied in the present work was isolated from the surface soils collected from Jiangsu Yancheng Wetland National Nature Reserve of Rare Birds. From three soils samples collected from the mudflat of *Spartina alterniflora* (33°36′ E, 120°36′ N), an identical strain was isolated and preserved in the HrLab (Halophyte Research Lab) of Nanjing University before the present study. The individual strains were denoted as *Mucoraceae* sp. SM-1 as the laboratory serial no. and finally named *A. jiangsuensis*. The holotype was preserved by China General Microbiological Culture Collection Center (CGMCC) and given the preservation No. CGMCC3.17001, simultaneously, lyophilized formats were stored at −20 °C in the HrLab of Nanjing University.

### 2.2. Morphological Studies and Growth Observation

The strain was subcultured on the following: modified Martin agar medium (MAM) comprising K_2_HPO_4_ (1 g), MgSO_4_·7H_2_O (0.5 g), NaCl (11.5 g), agar (20 g), peptone (5 g), glucose (10 g), gelose (10 g), and distilled water (1000 mL); PDA; Sabouraud dextrose agar (SDA) comprising glucose (4 g), peptone (10 g), agar (20 g), NaCl (11.5 g), and distilled water (1000 mL); Czapek agar (CZA, Difco, Becton Dickinson, France); millet extract agar (MEA) comprising millet extract (10 g), agar (20 g), NaCl (11.5 g), and distilled water (1000 mL); and oat medium agar (OMA) comprising oat meal (17 g), agar (20 g), NaCl (11.5 g), glucose (2 g), sucrose (3 g), and distilled water (1000 mL). The strain was incubated in the dark at 28 °C for 9 days. Superficial characteristics, color, edge feature, colony diameter, and time cost for the full coverage of the fungal colonies were documented daily. Three replicates were made in each test.

### 2.3. Optimum Growth Temperature and Salt Concentration

The strain was subcultured on MAM at 10 °C, 20 °C, 24 °C, 28 °C, 32 °C, 37 °C, and 42 °C then cultured in the dark at for 4 days. The strain was grown on Martin broth medium comprising K_2_HPO_4_ (1 g), MgSO_4_·7H_2_O (0.5 g), peptone (5 g), glucose (10 g), gelose (10 g), and distilled water (900 mL), to which 0.8%, 1.0%, 1.15%, 1.3%, 1.5%, and 1.8% NaCl was added. The inoculation dose was 2%. Then, the strain was placed on a shaker in the dark for 72 h at 180 rpm and 28 °C. The dry weight of mycelium was recorded. The salinity condition that yielded the maximum biomass was selected as the salinity condition for the subsequent experiments. Three replicates were made in each test. 

### 2.4. The Specific Method of Inducing Sporulation

*Apophysomyces* species failed to sporulate on routine mycological media [19]. To reliably and consistently obtain sporulation, the *A. jiangsuensis* was cultured in a wide-mouthed glass bottle on millet and bran solid medium (MBM) comprising millet (7.5 g), two drops of sesame oil, and 15 mL of liquid. The liquid consisted of yeast extract (0.5 g), sodium tartrate (0.1 g), sodium glutamate (0.1 g), FeSO_4_·7H_2_O (0.01 g), and distilled water (15 mL). Distilled coverslips were embedded in a 45° angle into MBM inoculated with the fungal strain. Coverslips were placed perpendicularly to the trace line of inoculation. Wide-mouthed glass bottles were then cultured at 28 ± 1 °C until the mycelium grew and covered the coverslips, which were carefully processed for observation with an optical microscope and photographed. When mycelia on the coverslips began to turn yellow, they were examined by scanning electron microscopy (SEM) and optical microscopy.

### 2.5. DNA Extraction, Amplification, and Sequencing

DNA was extracted and purified directly from fungal colonies following a slightly modified fast DNA kit protocol (Bio101, Vista, CA, USA) that consisted of a homogenization step repeated thrice with a Fast Prep FP120 instrument (Thermo Savant, Holbrook, NY, USA). DNA was quantified by the GeneQuant pro (Amersham Pharmacia Biotech, Cambridge, UK). The internal transcribed spacer (ITS) region of the nuclear rDNA was amplified using the primer pair ITS5-ITS4 and the D1–D2 domains of the 28S rDNA gene were amplified using the primer pair NL1–NL4 (Zhang and Su 2006). The PCR mixture (25 μL) contains 1.5 μL of template DNA, 10 mM Tris–HCl (pH 8.3), 50 mM KCl, 2.5 mM MgCl_2_ (10× Perkin-Elmer buffer II plus MgCl_2_ solution Roche Molecular Systems, Branchburg, NJ, USA), 100 mM of each dNTP, 1 mM of each primer, and 1.5 U of Ampli Taq DNA polymerase (Roche). The amplification program for the three DNA fragments included an initial denaturation at 94 °C for 5 min, followed by 28 cycles of denaturation at 95 °C for 30 s, annealing for 1 min at 54 °C, and extension for 1 min at 72 °C, ending with 5 min at 72 °C. The products were purified with an Illustra GFXTM PCR DNA and Gel Band Purification Kit (General Electric Healthcare, Buckingham shire, UK) and stored at −20 °C until sequencing. PCR products were sequenced using the same primers used for amplification and following the Taq Dye Deoxy Terminator Cycle Sequencing Kit protocol (Applied Biosystems, Gouda, The Netherlands). Reactions were run on a 310 DNA sequencer (Applied Biosystems, Gouda, The Netherlands). Consensus sequences were obtained using the Auto assembler program (Perkin-Elmer-Applied Biosystems) and Seqman software (Laser-gene, Madison, WI, USA).

### 2.6. Phylogenetic Analysis

Multiple sequence alignments with reference sequences of *Apophysomyces* strains were performed using CLUSTAL X 2.1 [20]. Details of the sequences used for phylogenetic analysis are provided in Table 1. The alignment was checked and manually modified in GENEDOC [21] to remove the extra 5′ and 3′ sequences from where the sequences were overlapped. The combined alignments were deposited in TreeBase (http://treebase.org/treebase-web/home.html, submission ID: 27044). Phylogenetic analyses were conducted by Bayesian inference (BI) and maximum likelihood (ML) analyses using MrBayes v.3.2.6 [22] and IQ-TREE v.1.6.3 [23], respectively. jModelTest 2.1.10 [24] was used to compare the likelihood of the different nested models of DNA substitution and to select the best-fit model for the dataset through Akaike information criterion (AIC). BI analysis was performed using a Markov chain Monte Carlo (MCMC) algorithm. There were 5,000,000 generations that resulted in 50,000 trees for sampling every 100 generations. The first 12,500 trees (the first 25% was sampled by default in the software) were discarded as the burn-in phase, and the remaining 37,500 trees were used to calculate posterior probabilities (PP) values in the majority rule consensus tree. In ML analysis, we obtained branch supports with the ultrafast bootstrap [25] implemented in the IQ-TREE. The pairwise genetic distances (p–distance) were calculated using MEGA version 6 [26] with substitutions: Transitions + Transversions.

### 2.7. Dissolved Phosphorus Ability

Liquid medium formulation: glucose 10 g, (NH_4_)_2_SO_4_ 0.5 g, MgSO_4_·7H_2_O 0.3 g, NaC1 11.3 g, KCl 0.3g, FeSO_4_·7H_2_O 0.03 g, MnSO_4_·7H_2_O 0.03 g, and distilled water 1000 mL. Four phosphorus sources were added to the experimental group, as follows: (1) Ca_3_(PO_4_)_2_ at 5 g; (2) lecithin at 0.6 g and CaCO_3_ at 5 g; (3) AlPO_3_ at 4 g; and (4) FePO_4_·4H_2_O at 7.2 g, respectively. They were respectively placed in triangle bottles with the liquid content of 50/250 mL, and the inoculating amount was 2%. Then, these bottles were placed on a rotary shaker at 180 rpm and at 28 °C for 4 days. The supernatant was collected by centrifugation, and the available phosphorus content was determined by the molybdenum blue coloration method [27]. Three replicates were made in each test. The difference between the experimental group and the corresponding control group was determined using Microsoft Excel.

### 2.8. Low Molecular Weight (LMW) Organic Acids Derived from the Culture Liquor

To test the evolution of the pH of the fermented fungal suspension liquor, a modified liquid Martin medium was prepared with the addition of 1.15% NaCl. The initial inoculum dosage was set up at 5% (v/v) to a volume of 150 mL in 250 mL containers, which was then incubated at 28 ± 1 °C at a rotation speed of 160 rpm. The pH was measured every 24 h with triplicates for the treated and control groups.

The GC-TOF-MS approach was applied to quantify the major LMW organic acids (oxalic, succinic, and malic acids) produced by *A. jiangsuensis* SM-1 out of the fermentation liquor. After 7 days of incubation, the suspended mycelia were filtered prior to the methyl esterification and extraction of three acids. Preparation of standard solutions: benzoic acid standard solution was prepared by adding 2 g benzoic acid into 20 mL methanol; succinic, oxalic, and malic acid standard solutions were prepared by dissolving 2 g oxalate acid, 0.1 g succinic acid, and 0.1 g malic acid and diluting with water (250 mL). Methyl esterification and extraction: the filtered test solution and all standard solutions were mixed with 1 mL benzoic acid and 50 mL of 10% sulfuric acid-methanol (*v*/*v*) solution, and the mixtures were shaken at 30 °C for 24 h at 160 rpm. Mixtures were then extracted thrice with 30 mL of CH_2_Cl_2_ each time, and the extractions were pooled and washed with 50 mL of saturated NaCl solutions twice. The extracted liquors were mixed with 25 g Na_2_SO_4_ and left over the night before filtration. The solvent of filtered solutions was removed with the rotary evaporator, and then, the residues were dissolved with 5 mL of CH_2_Cl_2_ for GC analysis. TOFMS conditions: EI source electron energy of 70 eV, electron multiplier voltage of 1976 V, the quality of the scan range was 30–550 m/z, ion source temperature of 210 °C, quadrupole temperature of 150 °C, and the collected spectra were retrieved with the Nist02 library. GC conditions: column: HP-5C (30 m × 0.25 mm × 0.25 µm); injection volume: 1 µL, split (10:1); inlet temperature: 250 °C; carrier gas: N_2_; programmed temperature at 40 °C holding 2 min, and then raised at a rate of 5 °C/min to 240 °C, maintained 2 min, detector: FID; detector temperature: 300 °C.

## 3. Results

### 3.1. Phylogenetic Analyses

The topologies of each single-gene and the two-gene (ITS, 28S rDNA) trees were similar. Therefore, we show only the two-gene combined tree. According to the phylogenetic result, the *Apophysomyces* genus was separated into four main clades with high statistical support. Clade 1 (ML/BI = 100/1) contains *A. variabilis* and *A. elegans*, whereas clade 2 (ML/BI = 89/0.96) was clustered with *A. trapeziformis*, *A. mexicanus*, and *A. ossiformis*. *A. thailandensis* was assigned to clade 3 (ML/BI = 100/1). *A. jiangsuensis* was clearly separated from the other *Apophysomyces* species and formed an independent lineage (clade 4) with high statistical support (ML/BI = 100/1). *A. jiangsuensis* is phylogenetically recognized as a novel species (Figure 1).

The ITS–28S rDNA combined genetic distance between *A. jiangsuensis* and other *Apophysomyces* species ranged from 8% to 12.10%. This genetic distance was more than 3%, which indicated that this fungus is a new species [28].

### 3.2. Taxonomy

*Apophysomyces jiangsuensis* Siyu Li and Pei Qin, sp. nov.

MycoBank No.: MB 837380.

Etymology: from ‘*jiangsuensis*’, which refers to Jiangsu, where the soil containing the new fungus was collected.

Holotype: China, Jiangsu Yancheng Wetland National Nature Reserve of Rare Birds, mudflats of *Spartina alterniflora* (33°36′ E, 120°36′ N), 1 April 2009, Ruiming Han. (Holotype—CGMCC3.17001).

Gene sequences (from holotype): KF048102 (ITS), KF055451 (28s rDNA).

Diagnosis: Differs from other *Apophysomyces* species in having spherical rather than pyriform sporangia and the spore wall is ornamented rather than smooth.

Description: Colonies were fast growing on MAM and PDA medium with 1.15% NaCl, flocculent mycelium. The color of the strain was creamy on top and whitish-brown underneath, the edges were smooth and had no odor, and the reverse was brown (Figure 2a,b). On the MEA medium, the strain had the aroma of mushrooms. On OMA and SDA media, the growth was poor, and the edges were uneven, showing significant nutritional deficiencies. It did not grow on CZA medium (Table 2). Sporangiophores were erect, arising singly, unbranched and white with lengths of 40–100 µm, and the top of these sporangiophores formed the apophyses (Figure 2c). Under conditions with sufficient nutrients, 1–5 mycelia were produced, while under nutrient-deficient conditions, spores were produced (Figure 2d). Branching hyphae often had a septate basal segment resembling the “foot cell” (Figure 2e). False roots were thin-walled, white, and predominantly unbranched. Sporangia were multispored, small, nearly sphere, peach-shaped with diameters of 10–17 µm, when mature with distinct apophyses (Figure 2c). Spores formed on the top of these apophyses and were ellipsoidal with diameters of 4–5 × 6–7 µm and had raised white nubbins on surface (Figure 2f). The strain grew well in the salinity range of 0.8%–1.3% (Figure 3) and good growth at 28 °C, but growth was sparse and slow at 20 °C, 24 °C, 32 °C, and 37 °C and hardly grew at 10 °C and 42 °C.

Notes: The colony of *A. jiangsuensis* isolated from salt marsh had seemingly identical characteristics to those from species of the previous genus *Apophysomyces* [1], which possessed false roots, flocculent mycelia, and a small amount of diaphragm. The colony appeared white first before turning brown. It had difficulty forming spores in a normal medium. A significant difference existed in microscopic structures. The sporangia of known *Apophysomyces* species were typically pyriform in shape and spores were smooth-walled, whereas the sporangia of *A. jiangsuensis* were nearly spherical, peach-shaped. The spores were ellipsoidal and had raised white nubbins on the surface. As with other species of *Apophysomyces*, *A. jiangsuensis* failed to sporulate on routine mycological media. To stimulate the production of spores, a special culture was made using millet and bran solid medium (MBM) at 28 + 1 °C for 7 days.

### 3.3. The Phosphate-Solubilizing Ability

In comparison with the control group, *A. jiangsuensis* cultured in liquid medium with 1.15% NaCl had an obvious decomposition effect on three kinds of phosphorus sources, namely, lecithin, Ca_3_(PO_4_)_2_, and AlPO_3_; and the available phosphorus concentrations were 16.60, 13.60, and 15.50 mg/L, respectively. However, it had no obvious decomposition effect on FePO_4_·4H_2_O (Figure 4).

### 3.4. pH Dynamics of the Culture Liquor

During the 7-day culture of *A. jiangsuensis*, the pH of the fermented liquid progressively declined from 6.85 at the beginning to 2.27 at the end, compared with 6.45 in the medium without inoculation (Figure 5). A rapid decline was observed from days 2 to 4, whereas a significant difference existed between the control and inoculated groups (*p* < 0.0l), indicating a strong effect of the strain on the medium pH value.

### 3.5. LMW Organic Acids Derived from the Culture Liquor

TOFMS analysis identified that the LMW organic acid species collected in the culture liquor included oxalic, succinic, and malic acids and a trace of citric acid. According to the standard curve regression equation, the concentrations of the measurable organic acids were as follows: oxalic acid, 652.50 mg/L; succinate, 9.14 mg/L; and malic acid, 3.16 mg/L, respectively (Figure 6). When mixing standard acid samples at the above concentrations with non-inoculated medium, the pH of the medium was 2.88 on average.

## 4. Discussion

A newly isolated phosphate-solubilizing fungus from the topsoil of *S. alterniflora* habitats in Yancheng coastal salt marsh was identified by morphological characteristics and phylogenetic analyses. Colony and spores’ morphologies are the major bases for the taxonomic identification of fungi in this group. Morphologically, the *A. jiangsuensis* colony shows identical features with those from the genus *Apophysomyces*, which has been described previously, based on naked eye observations [1]. However, significant differences in sporangia and spores were observed on a microscopic scale. The sporangia of known *Apophysomyces* species are typically pyriform in shape and spores are smooth-walled [9]. By contrast, the sporangia of *A. jiangsuensis* are nearly spherical, peach-shaped. The spores are ellipsoidal and have raised white nubbins on the surface. As with other species of *Apophysomyces*, *A. jiangsuensis* failed to sporulate on routine mycological media, on sterilized distilled water [19], and on Sabouraud dextrose agar. It was transferred to plates containing sterile distilled water with 0.2 mL of 10% yeast extract solution that was filter sterilized [29]. To stimulate the production of spores, a special culture was performed on MBM solid medium at 28 + 1 °C for 7 days.

Bonifaz et al. (2014) divided *Apophysomyces* into two main clades using a phylogenetic tree derived from maximum likelihood analysis of a combined ITS, 28S rDNA, and histone 3 (H3) genes and the described a novel species *A. mexicanus*, which was placed between *A. ossiformis* and *A. trapeziformis* [10]. Khuna et al. (2019) separated *Apophysomyces* into three main clades, and *A. thailandensis* formed the clade III as a novel species [11]. In the current study, the topology of the phylogenetic tree was similar to that described in Bonifaz et al. (2014) and Khuna et al. (2019) [10,11], although only ITS and 28S rDNA sequences were used to construct the current phylogenetic tree. Moreover, *A. jiangsuensis* formed a distinct clade in this study as in previous studies and was at a great genetic distance from other *Apophysomyces* species, which clearly indicated that this fungus is a novel species.

The productivity of coastal tidal flat vegetation was believed to be limited by nitrogen, but the growth of the nitrogen-fixing microorganism in plant roots was found to be limited by phosphorus. Therefore, increasing the available phosphorus content in coastal tidal flat soil would effectively increase the nitrogen supply of plants and thus increase the crop yield [30]. *A. jiangsuensis* grows well in a salinity range of 0.8%–1.3% and can effectively improve the available phosphorus content, because it can decompose inorganic phosphorus compounds and organophosphorus compounds, such as lecithin, Ca_3_(PO_4_)_2_, and AlPO_3_. Thus, it is an excellent phosphorus solubilizer for salinity. This finding is similar to those obtained by previous studies on *A. thailandensis*, which can solubilize different insoluble minerals (Ca, Co, Fe, Mn, Cu, and Zn-containing minerals) [18]. However, *A. jiangsuensis* has no obvious effect on the decomposition of FePO_4_·4H_2_O, which was possibly because the inorganic acids or organic acids produced by the fungus have low solubility in insoluble ferrite.

The fermentation process of PSMs tends to lower the pH level of the fermentation liquor. The rapid decline of pH during days 2–4 suggested the existence of a logarithmic growth phase period of *A. jiangsuensis*, during which the fungus experienced the highest rate of grow and division while organic acid secretion was maximized. The average pH at day 7 was low (2.27), showing that *A. jiangsuensis* created a conducive environment for survival under alkaline saline conditions and for it to become the dominant fungal species. Moreover, it decomposes the surrounding phosphates into free forms to promote plant growth. The undergoing mechanism can be simplified as follows: (1) The phosphorus absorbed by plants promotes the increase of the rhizospheric organic fertilizer from exudation and die-back of fine roots. (2) Organic fertilizer stimulates growth and reproduction of various PSMs. (3) Acidification of the microenvironment is elevated due to these microorganisms. (4) The solubilization of insoluble phosphates and other mineral nutrients was accelerated. Out of the LMW organic acids secreted by *A. jiangsuensis*, oxalic acid is the most considerable quantitatively. Oxalic acid can dissolve lecithin, Ca_3_(PO_4_)_2_, AlPO_3_, and others and is known to form complexation with or chelate with metal ions, such as aluminum, iron, calcium, and magnesium to release phosphorus [31]. Oxalic acid is one of the simplest organic dicarboxylic acids and is known for its superior ability to dissolve rock phosphate after citric acid. Thus, *A. jiangsuensis* nearly has the most efficient substance to neutralize alkaline substances and to mobilize phosphates in the surrounding substrate. Thus, it is especially suited to survive in and to ameliorate salt marsh habitats. The simulation of acidification effect using pure LMW acids resulted in a pH of 2.88, which was not far from the detected value of 2.27 in fermented liquor, indicating that *A. jiangsuensis*-secreted organic acids play an important role in P solubilization. The simulated condition had a pH value that was higher by 0.61 compared with fermentation liquor, which can be attributed to the putative incomplete esterification of organic acids during extraction and/or the existence of other undetected trace organic contents. These factors may lead to the lower measured contents of organic acids compared with the real values, or the fermentation process contained other auxiliary supplement H+ mechanism.

In conclusion, a combination of morphological characteristics and phylogenetic analysis strongly supports the proposal that the fungus is a new species. *A. jiangsuensis* excretes large amounts of organic acids and creates a conducive environment. On the one hand, it is especially suited to survive in and to ameliorate salt marsh habitats. On the other hand, it can decompose the surrounding phosphates into free forms to promote plant growth. Thus, it is an excellent phosphorus solubilizer for salinity. This study is of great importance for the development of a phosphorus solubilizer to ameliorate soil phosphorous nutrition and to improve the limited plant productivity in the tidelands.

## Figures and Tables

**Figure 1 microorganisms-08-01868-f001:**
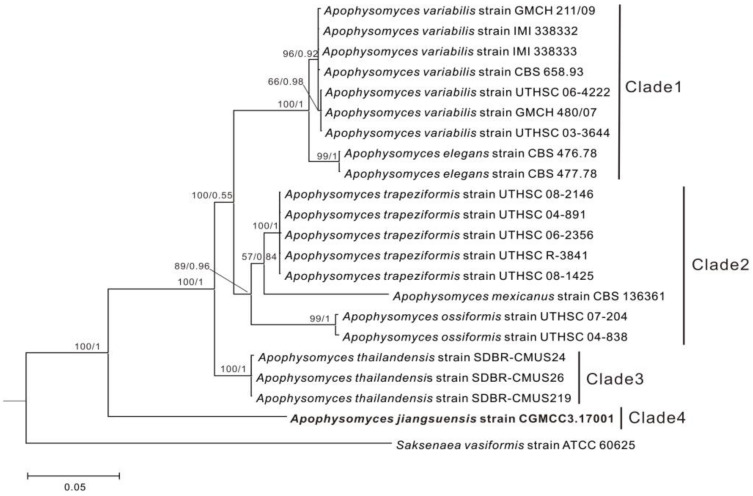
Phylogenetic tree derived from Bayesian inference (BI) and maximum likelihood (ML) analysis of a combined ITS and 28S rDNA sequences. *Saksenaea vasiformis* was used as an outgroup.

**Figure 2 microorganisms-08-01868-f002:**
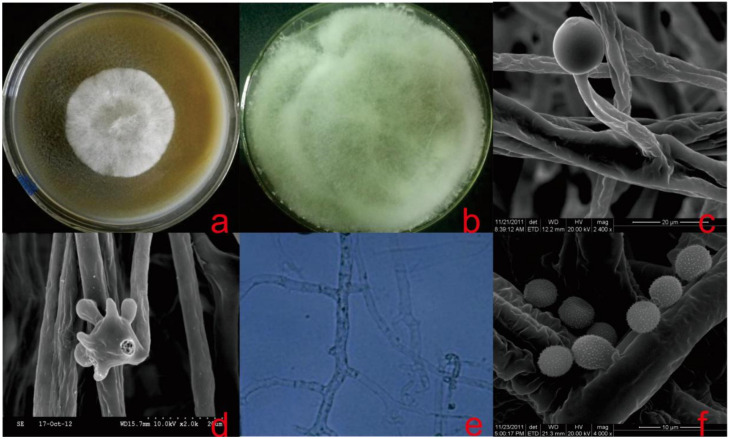
*Apophysomyces jiangsuensis*. (**a**) Fungal colony after 72 h on PDA; (**b**) covered the Petri dish after 5 days on MAM; (**c**) sporangiophores and sporangia; (**d**) forming spores and hyphae; (**e**) septum in the hypha; and (**f**) spores.

**Figure 3 microorganisms-08-01868-f003:**
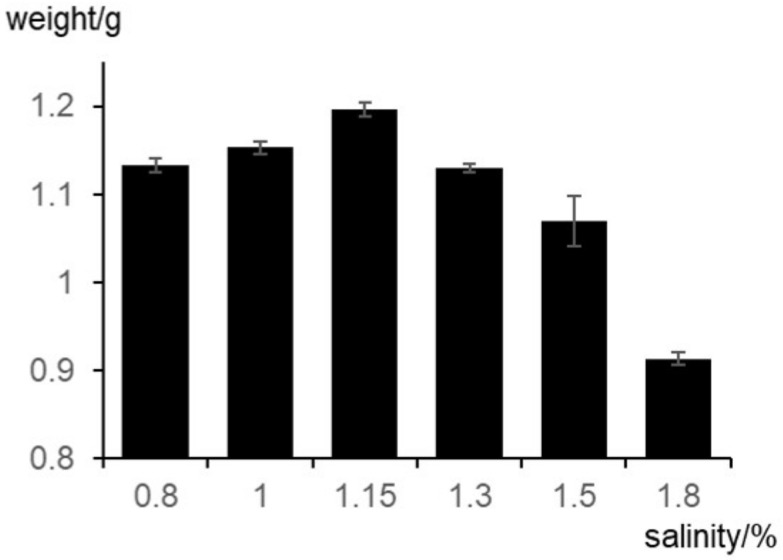
The effect of different salinity on biomass.

**Figure 4 microorganisms-08-01868-f004:**
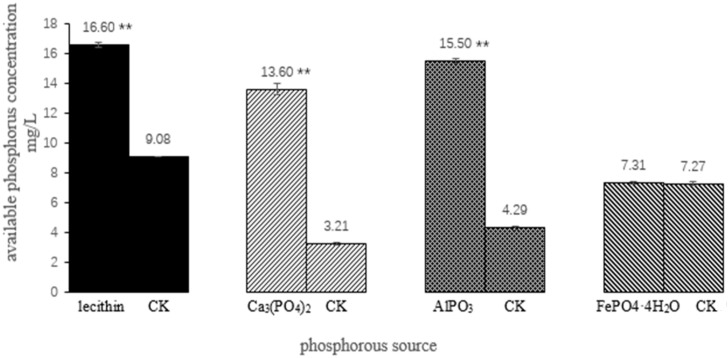
The ability of the strain to disintegrate phosphorus from different sources. ** showed significant difference between the treatment group and the control group after a *t* test (*p* < 0.01).

**Figure 5 microorganisms-08-01868-f005:**
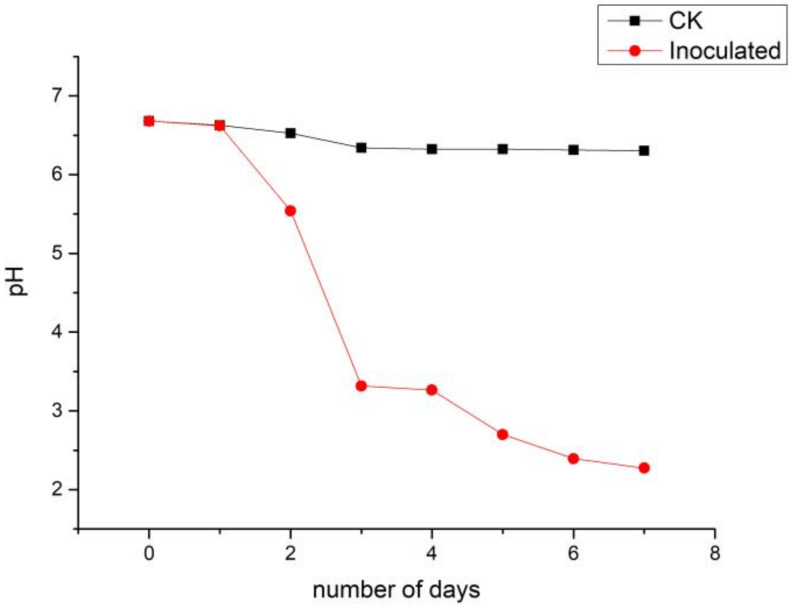
pH of the culture liquid during a 7-day incubation of *A. jiangsuensis* in the culture medium.

**Figure 6 microorganisms-08-01868-f006:**
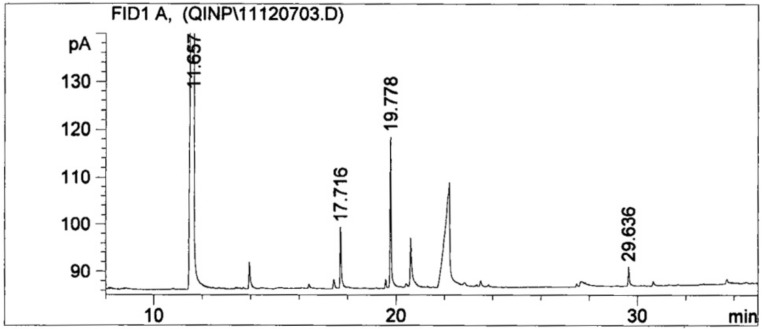
Sample organic acid methyl ester by gas chromatogram.

**Table 1 microorganisms-08-01868-t001:** The strains used in this study and their GenBank accession numbers.

Species	Isolate	Source	GenBank Accession No.		
			ITS	28S	Reference
*A. variabilis*	UTHSC 06-4222	Dolphin, Bahamas	FN556428	FN554255	[9]
*A. variabilis*	UTHSC 03-3644	Dolphin, FL, USA	FN556431	FN554259	[9]
*A. variabilis*	IMI 338333	Daly river, Australia	FN556439	FN554256	[9]
*A. variabilis*	IMI 338332	Ankle aspirate, Australia	FN556438	FN554257	[9]
*A. variabilis*	GMCH 480/07	Cutaneous infection, India	FN556442	FN554253	[9]
*A. variabilis*	GMCH 211/09	Cutaneous infection, India	FN556443	FN554254	[9]
*A. variabilis*	CBS 658.93	Osteomyelitis, Netherlands Antilles	FN556436	FN554258	[9]
*A. trapeziformis*	UTHSC R-3841	Necrotic face tissue, GA, USA	FN556434	FN554263	[9]
*A. trapeziformis*	UTHSC 08-2146	Skin biopsy, CO, USA	FN556430	FN554260	[9]
*A. trapeziformis*	UTHSC 08-1425	Abdominal tissue, PHA, USA	FN556429	FN554261	[9]
*A. trapeziformis*	UTHSC 06-2356	Dolphin, TX, USA	FN556427	FN554262	[9]
*A. trapeziformis*	UTHSC 04-891	Sinus, MN, USA	FN556433	FN554264	[9]
*A. thailandensis*	SDBR-CMUS26	Soil, Thailand	MH733251	MH733254	[11]
*A. thailandensis*	SDBR-CMUS24	Soil, Thailand	MH733250	MH733253	[11]
*A. thailandensis*	SDBR-CMUS219	Soil, Thailand	MH733252	MH733255	[11]
*A. ossiformis*	UTHSC 07-204	Facial cellulitis, AZ, USA	FN556435	FN554251	[9]
*A. ossiformis*	UTHSC 04-838	Cellulitis wound leg, MN, USA	FN556432	FN554252	[9]
*A. mexicanus*	CBS 136361	Human, Mexican	HG974255	HG974256	[10]
*A. elegans*	CBS 477.78	Soil, Gorakhpur, India	FN556437	FN554250	[9]
*A. elegans*	CBS 476.78	Soil, Deoria, India	FN556440	FN554249	[9]
*A. jiangsuensis*	CGMCC3.17001	Soil, JiangSu, China	KF048102	KF055451	This study

**Table 2 microorganisms-08-01868-t002:** Growth characteristics of *A. jiangsuensis* on different media.

Medium	Isolate Diameter (mm/day)				Isolate Color		Height (mm)	Odor
	1	3	5	7	Front	Reverse		
MAM	15.4 ± 0.07	80.6 ± 0.07	90 ± 0.00	90 ± 0.00	white	brown	77.3 ± 0.07	/
PDA	12.1 ± 0.03	70.4 ± 0.05	90 ± 0.00	90 ± 0.00	white	brown	70.1 ± 0.53	/
SDA	1.7 ± 0.00	35.3 ± 0.07	72.8 ± 0.09	90 ± 0.00	white	white	35.3 ± 0.10	/
MEA	5.3 ± 0.05	50.7 ± 0.12	84.4 ± 0.09	90 ± 0.00	white turn yellow	brown	120.4 ± 0.12	mushroom aroma
OMA	2.1 ± 0.05	20.4 ± 0.07	31.2 ± 0.07	33.3 ± 0.05	white turn yellow	brown	20 ± 0.47	/
CZA	–	–	–	–	–	–	–	/

MAM = modified Martin agar medium, PDA = potato dextrose agar medium, SDA = Sabouraud dextrose agar medium, MEA = millet extract agar medium, OMA = oat medium agar. “–” = no growth, CZA = Czapek agar medium. “/” = no odor.

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
