# Peer review of "Apophysomyces jiangsuensis sp. nov., a Salt Tolerant and Phosphate-Solubilizing Fungus from the Tidelands of Jiangsu Province of China"

_microorganisms, 2020, doi:10.3390/microorganisms8121868_

Round 1
Reviewer 1 Report
Here is the review of manuscript entitled "Apophysomyces jiangsuensis sp. nov., a salt tolerant and phosphate-solubilizing fungus from the tidelands of Jiangsu Province of China".
The object of the paper is phosphate-solubilizing fungus isolated from topsoil of Spartina alterniflora habitats in Yancheng (China) coastal salt marsh. The species is analysed on the basis of its morphology, cultural features, and molecular characters (ITS, 28S gene markers). The conclusion of the study was that it represents a new species in the genus Apophysomyces. So, the authors described it as a new species following the International Code of Nomenclature for algae, fungi, and plants (Shenzen, 2017). In addition, the authors studied optimum growth temperature and salinity, as well as phosphate-solubilizing ability of the fungus in the culture. In the liquid culture, the species had strong decomposition effect on three kinds of phosphorus sources, lecithin, Ca3(PO4)2, and AlPO3.
The paper was very well prepared. The research methods are suitable and well conducted. This paper contain important novel finding and I am recommending it for publication after the minor correction.
There are a few minor details in the text that need to be corrected.
So, please see the list of my remarks:
22 multilocus -> Delete the word "multilocus" since you analysed only two genes (ITS, 28S)
23 belonged -> belongs
36 The genus Apophysomyces (Comment: Please explain taxonomic position of the genus Apophysomyces (Saksenaeaceae, Mucorales, Mucoromycetes). Use some recent taxonomic review as reference, like Wijayawardene & al. 2020 - Outline of fungi and fungus like taxa. Mycosphere 11: 1060–1456.
42 Bonifaz -> Bonifaz et al.
56 Mucor fungus -> mucoroid fungus
59 was hampered -> were hampered
64-65 to propose a newly identified salt tolerant P-solubilizing fungus Apophysomyces jiangsuensis -> to describe salt tolerant P-solubilizing fungus Apophysomyces jiangsuensis as a new species
67 multilocus -> Delete the word "multilocus" since you have only two genes (ITS, 28S)
67-68 as well as the evaluation of morphological characteristics -> as well as to evaluate its morphological characteristics
76 reserved -> preserved
110 scanned -> examined
118 ul -> μL (twice in the same line)
148 in Table 1 change thailandensi -> thailandensis
195 more than 3% -> comment: cite reference which uses threshold of 3%
196 fungal -> fungus
196-197 in Figure 1 change thailandensi -> thailandensis
208 soils -> soil
210 reserved -> preserved
247 to the previously -> to those from species of the previously
251 spores formed -> spores were formed
286 fungi -> fungi in this group
286 with genus -> with those from the genus
294 culture -> media
307 fungal was -> fungus is
352 significance -> importance
Best,
Reviewer
Author Response
Point 1: Line36 : The genus Apophysomyces (Comment: Please explain taxonomic position of the genus Apophysomyces (Saksenaeaceae, Mucorales, Mucoromycetes). Use some recent taxonomic review as reference, like Wijayawardene & al. 2020 - Outline of fungi and fungus like taxa. Mycosphere 11: 1060–1456.
Response 1: The taxonomic position of the genus Apophysomyces is added.
In Line 36-42:“ The genus Apophysomyces was first proposed by Misra et al. (1979) and A. elegans was the type specimen that was isolated and identified in the soils in northern India. Apophysomyces belongs to the order Mucorales (Benny et al. 2001) and Benny et al. (2001) identified it as a genus in the family Mucoraceae based predominantly on morphology, while Hoffmann et al. (2013) put it into Saksenaeaceae based on a phylogenetic analysis of four molecular markers. Meanwhile, Wijayawardene et al. (2020) confirmed that Apophysomyces was contained in Saksenaeaceae on the basis of phylogeny of the kingdom Fungi.”
At the same time, heat resistance, habitat and pathogenicity is added.
In Line 45-50:“ And it is pretty thermotolerant that can grow rapidly between 26 and 42 ℃ (Cooter et al. 1990). Apophysomyces is usually found in soil, decaying vegetation, and detritus, which has been reported to cause severe human infections that is more common in tropical and subtropical regions (Misra et al. 1979; Chakrabarti et al. 2003). The type species A. elegans has been reported as an agent of zygomycosis in immunocompromised patients (Kimura et al. 1999; Reddy et al. 2008). ”
Point 2:195 more than 3% -> comment: cite reference which uses threshold of 3%
Response 2: It has been corrected.
Cite reference:“Nilsson RH, Kristiansson E, Ryberg M, Hallenberg N, Larsson KH (2008) Intraspecific ITS variability in the kingdom Fungi as expressed in the international sequence databases and its implications for molecular species identification. Evolutionary Bioinformatics 4: 193– 201. https://doi.org/10.4137/EBO.S653. ”
References
Benny, G.L.; Humber, R.A.; Morton, J.B. Zygomycota: Zygomycetes. In: McLaughlin DJ, McLaughlin EG, Lemke PA (eds), The mycota. Vol. VIIA. Systematics and evolution 2001, 113–146.
Hoffmann, K.; PawÅ‚owska, J.; Walther, G.; Wrzosek, M.; Hoog, de, G.S.; Benny, G.L.; Kirk, P.M.; Voigt, K. The family structure of the Mucorales: a synoptic revision based on comprehensive multigene-genealogies. Persoonia 2013, 30: 57–76.
Wijayawardene, N.N.; Hyde, K.D.; Al-Ani, L.K.T. Outline of Fungi and fungus-like taxa. Mycosphere 2020, 11:1060-1456.
Cooter, R.D.; Lim, I.S.; Ellis, D.H.; Leitch, I.O.W. Burn wound zygomycosis caused by Apophysomyces elegans. J. Clin Microbiol. 1990, 28:2151–3.
Chakrabarti, A.; Ghosh, A.; Prasad, G.S.; David, J.K.; Gupta, S. Apophysomyces elegans: an emerging zygomycete in India. J. Clin. Microbiol. 2003, 41: 783–788.
Kimura, M.; Smith, M.B.; McGinnis, M.R. Zygomycosis due to Apophysomyces elegans: report of 2 cases and review of the literature. Arch. Pathol. Lab. Med. 1999, 123: 386–390.
Reddy, I.S.; Rao, N.R.; Reddy, V.M.; Rao, R. Primary cutaneous mucormycosis (Zygomycosis) caused by Apophysomyces elegans. Indian J. Dermatol. Venereol. Leprol. 2008, 74: 367–370.
Nilsson R,H,; Kristiansson, E.; Ryberg, M.; Hallenberg, N.; Larsson K.H. Intraspecific ITS variability in the kingdom Fungi as expressed in the international sequence databases and its implications for molecular species identification. Evol Bioinform 2008, 4: 193– 201.
The following minor mistakes were corrected accordingly.
21 multilocus -> Delete the word "multilocus" since you analysed only two genes (ITS, 28S)
22 belonged -> belongs
52 Bonifaz -> Bonifaz et al.
65 Mucor fungus -> mucoroid fungus
68 was hampered -> were hampered
74-75 to propose a newly identified salt tolerant P-solubilizing fungus Apophysomyces jiangsuensis -> to describe salt tolerant P-solubilizing fungus Apophysomyces jiangsuensis as a new species
77 multilocus -> Delete the word "multilocus" since you have only two genes (ITS, 28S)
77-78 as well as the evaluation of morphological characteristics -> as well as to evaluate its morphological characteristics
86 reserved -> preserved
122 scanned -> examined
130 ul -> μL (twice in the same line)
160 in Table 1 change thailandensi -> thailandensis
209 fungal -> fungus
210 in Figure 1 change thailandensi -> thailandensis
217 soils -> soil
250 to the previously -> to those from species of the previously
288 fungi -> fungi in this group
288-289 with genus -> with those from the genus
294 culture -> media
307 fungal was -> fungus is
352 significance -> importance
Yours sincerely,
Siyu Li
Reviewer 2 Report
Overall the paper is fine and the topic worth publishing on, but the taxonomy section needs to be modified for your new species to be validly described.
For valid publication, an individual specimen needs to be cited as the holotype. Although a living culture is allowed as a holotype, this is not ideal and if it were possible to deposit a dried culture from CGMCC3.17001 in a fungarium, that would be better as a holotype. This could then be cited as (Holotype – [fungarium acronym and number], ex-type culture CGMCC3.17001).
If you decide to keep the living culture as the holotype, you need to state explicitly that the culture is preserved in a metabolically inactive state (and state how this is achieved, either it is lyophilised or it is stored in liquid N). This statement could be included at the end of Section 2.1.
Your diagnosis is a description and the text here is very strange. For examples of descriptions of these fungi, look over the description in Misra et al. 1979 and https://mycology.adelaide.edu.au/descriptions/zygomycetes/apophysomyces/
The detail in your description about arrangement of the hyphae during growth does not seem relevant, whereas the details mentioned in the Notes section should be in the description. What appear to be important features that are mention in the ‘Notes’ section such as ‘false roots’, ‘follicles’ and ‘diaphragm’ are not mentioned in the description. Based on your notes, they are key morphological characters for this genus and as such should be in the description. Is ‘follicle’ the same as ‘apophyses’ as used in the Misra et al. (1979) description? If so, best to use the same term that they use.
The taxonomy section could be set out something like this
3.2. Taxonomy
Apophysomyces jiangsuensis Siyu Li & Pei Qin, sp. nov.
MycoBank No.: MB 837380
Etymology: from ‘jiangsuensis’, which refers to Jiangsu, where the soil containing the new fungus was collected.
Holotype: China, Jiangsu Yancheng Wetland National Nature Reserve of Rare Birds, mudflats of Spartinia alterniflora (33°36′E,120°36′N), 1 April 2009, [name of the collector] (Holotype – CGMCC3.17001).
Gene sequences (from holotype): KF048102 (ITS), KF055451 (28s rDNA).
Diagnosis: Differs from other Apophysomyces species in having spherical rather than bell-shaped or funnel-shaped apophyses on the sporangia and the spore wall is ornamented rather than smooth.
Description: [full description including details on spore size, sporangia, appearance in culture (colour, odour, etc), growth rates, etc. Is all the detail about hyphal branching patterns etc. useful for identifying this species? If not, cut all this out of the description.]
Notes: …
A few other specific comments:
Table 2 is not needed. The phylogeny in Fig 1 clearly shows your new species is phylogenetically distinct.
Fig. 2 caption – spell out Apophysomyces
Fig 2a – is this a MAM colony after less than 5 days growth? or growth on some other agar?
Table 3 – ‘brown’ not ‘brownness’; ‘mushroom aroma’ not ‘mushrooms’ aroma’
line 245 - ‘base of spore collapsed’ – is this not an SEM artifact?
Line 248 - is 'sallow' the same as 'brown'?
Author Response
Point 1:Are the methods adequately described? Can be improved?
Response 1: The methods has been slightly improved accordingly. Mainly added holotype storage method and modify some improper expression. Meanwhile, the results and discussion sections have also been improved.
Point 2:For valid publication, an individual specimen needs to be cited as the holotype. Although a living culture is allowed as a holotype, this is not ideal and if it were possible to deposit a dried culture from CGMCC3.17001 in a fungarium, that would be better as a holotype. This could then be cited as (Holotype – [fungarium acronym and number], ex-type culture CGMCC3.17001).
If you decide to keep the living culture as the holotype, you need to state explicitly that the culture is preserved in a metabolically inactive state (and state how this is achieved, either it is lyophilised or it is stored in liquid N). This statement could be included at the end of Section 2.1.
Response 2: The holotype storage is added at the end of Section 2.1 in Line 88-90:“The holotype was preserved by China General Microbiological Culture Collection Center (CGMCC) and given the preservation No. CGMCC3.17001, simultaneously, lyophilised formats were stored at -20 ℃ in the HrLab of Nanjing University.”
Point 3:Your diagnosis is a description and the text here is very strange. For examples of descriptions of these fungi, look over the description in Misra et al. 1979 and https://mycology.adelaide.edu.au/descriptions/zygomycetes/apophysomyces/
Response 3: Diagnosis has been improved to more direct in line 230-231.
“Diagnosis: Differs from other Apophysomyces species in having spherical rather than pyriform sporangia and the spore wall is ornamented rather than smooth.”
the methods has been slightly improved accordingly
Point 4:The detail in your description about arrangement of the hyphae during growth does not seem relevant, whereas the details mentioned in the Notes section should be in the description. What appear to be important features that are mention in the ‘Notes’ section such as ‘false roots’, ‘follicles’ and ‘diaphragm’ are not mentioned in the description. Based on your notes, they are key morphological characters for this genus and as such should be in the description. Is ‘follicle’ the same as ‘apophyses’ as used in the Misra et al. (1979) description? If so, best to use the same term that they use.
Response 4: In the description section, we deleted the description of unrelated mycelium growth and arrangement and added description of ‘false roots’, ‘apophysis’ and‘diaphragm’. ‘Follicle’is the same as ‘apophyses’, so we use the same term as in the Misra et al. (1979) description.
Point 5:Description: [full description including details on spore size, sporangia, appearance in culture (colour, odour, etc), growth rates, etc. Is all the detail about hyphal branching patterns etc. useful for identifying this species? If not, cut all this out of the description.]
Response 5: In the description section, we deleted the description of unrelated mycelium growth and arrangement and this part of the content has been corrected in line 234-248.
“Description:Colonies are fast growing on MAM and PDA medium with 1.15% NaCl, flocculent mycelium. The color of the strain is creamy on top and whitish-brown underneath, the edges are smooth and have no odor, and the reverse are brown (Figure 2a, b). On MEA medium, the strain has the aroma of mushrooms. On OMA and SDA media, the growth is poor, and the edges are uneven, showing significant nutritional deficiencies. It does not grow on CZA medium (Table 2). Sporangiophores are erect, arising singly, unbranched and white with lengths of 40–100 µm, and the top of these sporangiophores formed the apophyses (Figure 2c). Under conditions with sufficient nutrients, 1–5 mycelia are produced, while under nutrient-deficient conditions, spores are produced (Figure 2d). Branching hyphae often have a septate basal segment resembling the“foot cell”(Figure 2e). False roots are thin-walled, white and predominantly unbranched. Sporangia are multispored, small, nearly sphere, peach-shaped with diameters of 10–17 µm, when mature with distinct apophyses (Figure 2c). Spores formed on the top of these apophyses and are ellipsoidal with diameters of 4–5 × 6–7 µm and have raised white nubbins on surface (Figure 2f). The strain grows well in the salinity range of 0.8% to 1.3% (Figure 3) and good growth at 28°C, but growth is sparse and slow at 20°C, 24°C, 32°C, and 37°C and hardly grow at 10°C and 42°C.”
The taxonomy section is corrected as follows:
3.2. Taxonomy
Apophysomyces jiangsuensis Siyu Li & Pei Qin, sp. nov.
MycoBank No.: MB 837380
Etymology: from ‘jiangsuensis’, which refers to Jiangsu, where the soil containing the new fungus was collected.
Holotype: China, Jiangsu Yancheng Wetland National Nature Reserve of Rare Birds, mudflats of Spartinia alterniflora (33°36′E,120°36′N), 1 April 2009, Ruiming Han. (Holotype – CGMCC3.17001).
Gene sequences (from holotype): KF048102 (ITS), KF055451 (28s rDNA).
Diagnosis: Differs from other Apophysomyces species in having spherical rather than pyriform sporangia and the spore wall is ornamented rather than smooth.
Description:Colonies are fast growing on MAM and PDA medium with 1.15% NaCl, flocculent mycelium. The color of the strain is creamy on top and whitish-brown underneath, the edges are smooth and have no odor, and the reverse are brown (Figure 2a, b). On MEA medium, the strain has the aroma of mushrooms. On OMA and SDA media, the growth is poor, and the edges are uneven, showing significant nutritional deficiencies. It does not grow on CZA medium (Table 2). Sporangiophores are erect, arising singly, unbranched and white with lengths of 40–100 µm, and the top of these sporangiophores formed the apophyses (Figure 2c). Under conditions with sufficient nutrients, 1–5 mycelia are produced, while under nutrient-deficient conditions, spores are produced (Figure 2d). Branching hyphae often have a septate basal segment resembling the“foot cell”(Figure 2e). False roots are thin-walled, white and predominantly unbranched. Sporangia are multispored, small, nearly sphere, peach-shaped with diameters of 10–17 µm, when mature with distinct apophyses (Figure 2c). Spores formed on the top of these apophyses and are ellipsoidal with diameters of 4–5 × 6–7 µm and have raised white nubbins on surface (Figure 2f). The strain grows well in the salinity range of 0.8% to 1.3% (Figure 3) and good growth at 28°C, but growth is sparse and slow at 20°C, 24°C, 32°C, and 37°C and hardly grow at 10°C and 42°C.
Notes: Colony of A. jiangsuensis isolated from salt marsh had seemingly identical characteristics to those from species of the previously genus Apophysomyces [1, 21], which possessed false roots, flocculent mycelia, and small amount of diaphragm. The colony appear white first before turning brown. It has difficulty forming spores in a normal medium. Significant difference existed in microscopic structures. The sporangia of known Apophysomyces species are typically pyriform in shape and spores are smooth-walled, whereas the sporangia of A. jiangsuensis are nearly spherical, peach-shaped. The spores are ellipsoidal and have raised white nubbins on the surface. As with other species of Apophysomyces, A. jiangsuensis failed to sporulate on routine mycological media. To stimulate the production of spores, a special culture was made using millet and bran solid medium (MBM) at 28+1°C for 7 days.
Point 6:Table 2 is not needed. The phylogeny in Fig 1 clearly shows your new species is phylogenetically distinct.
Response 6:It has been deleted.
Point 7:Fig. 2 caption – spell out Apophysomyces
Response 7 Line 224: It has been corrected.
Point 8:Fig 2a – is this a MAM colony after less than 5 days growth? or growth on some other agar?
Response 8: Line 224: Fig 2a is PDA colony after 72h growth. It has been corrected.
Point 9:Table 3 – ‘brown’ not ‘brownness’; ‘mushroom aroma’ not ‘mushrooms’ aroma’
Response 9: Line 227:They have been corrected.
Point 10:line 245 - ‘base of spore collapsed’ – is this not an SEM artifact?
Response 10: We agree with the reviewer that base of spore collapsed is likely to have been formed during the SEM process, which we have removed.
Point 11:Line 248 - is 'sallow' the same as 'brown'?
Response 11: Line 252: Yes, it is, so we use the same term as 'brown' to correct.
Yours sincerely,
Siyu Li